**Data Availability Statement:** All relevant data are within the manuscript and its Supporting Information files.

**Funding:** The author(s) received no specific funding for this work.

# YouTube as an information source for bleeding gums: A quantitative and qualitative analysis

Jiali Wu[1], Danlin Li[2], Minkui Lin[1,2]*

1 Fujian Key Laboratory of Oral Diseases, Fujian Provincial Engineering Research Center of Oral Biomaterial, Stomatological Key Lab of Fujian College and University, School and Hospital of Stomatology, Fujian Medical University, Fujian, China, 2 Research Center of Dental Esthetics and Biomechanics, Fujian Medical University, Fujian, China

* linmk105@sina.com

## Abstract

Gum bleeding is a common dental problem, and numerous patients seek health-related information on this topic online. The YouTube website is a popular resource for people searching for medical information. To our knowledge, no recent study has evaluated content related to bleeding gums on YouTube™. Therefore, this study aimed to conduct a quantitative and qualitative analysis of YouTube videos related to bleeding gums. A search was performed on YouTube using the keyword "bleeding gums" from Google Trends. Of the first 200 results, 107 videos met the inclusion criteria. The descriptive statistics for the videos included the time since upload, the video length, and the number of likes, views, comments, subscribers, and viewing rates. The global quality score (GQS), usefulness score, and DISCERN were used to evaluate the video quality. Statistical analysis was performed using the Kruskal–Wallis test, Mann–Whitney test, and Spearman correlation analysis. The majority (n = 69, 64.48%) of the videos observed were uploaded by hospitals/clinics and dentists/specialists. The highest coverage was for symptoms (95.33%). Only 14.02% of the videos were classified as "good". The average video length of the videos rated as "good" was significantly longer than the other groups (p <0.05), and the average viewing rate of the videos rated as "poor" (63,943.68%) was substantially higher than the other groups (p <0.05). YouTube videos on bleeding gums were of moderate quality, but their content was incomplete and unreliable. Incorrect and inadequate content can significantly influence patients' attitudes and medical decisions. Effort needs to be expended by dental professionals, organizations, and the YouTube platform to ensure that YouTube can serve as a reliable source of information on bleeding gums.

## 1. Introduction

Bleeding gums are a common dental problem for people in their daily lives. Patients frequently present to the dental clinic with bleeding gums when brushing their teeth or blood in their

**Competing interests:** The authors have declared that no competing interests exist.

mouth in the morning after waking up [1]. Bleeding gums are mainly triggered by periodontal disease and occasionally by peri-implant diseases, direct trauma, viruses, fungal or bacterial infections, medications, pregnancy, dermatoses, and systemic disorders [2–10]. One of the earliest clinical signs of periodontal disease is bleeding gums [11]. According to the new classification of periodontal and peri-implant diseases and conditions for 2017 [12], periodontal diseases and conditions were separated into the following categories: 1. periodontal health, gingival diseases and conditions; 2. periodontitis; and 3. other conditions affecting the periodontium. Gingivitis is considered reversible, but without treatment, it can progress to periodontitis [13]. Severe periodontitis is the sixth most widespread disease worldwide and affects approximately 10.8% of the global population [14, 15]. It can result in tooth loss and negatively influences chewing ability, general health, and quality of life [16, 17]. Therefore, bleeding gums caused by periodontal disease should be treated immediately. However, according to a poll, only 2.8% of adults with symptoms of bleeding gums seek professional dental care, while 60% disregard the condition [18]. Although the current epidemic of COVID-19 has hindered travel and medical treatment, it should not delay people's decisions to seek medical attention, which can accelerate the progression of periodontal disease. The importance of educating patients about oral health cannot be overstated.

Health advice is traditionally sought from dental clinicians. The internet is a convenient and accessible source of health advice, and patients can ask questions anytime and anywhere without experiencing psychological stress. Internet use can reduce transmission rates by decreasing contact, particularly during the COVID-19 outbreak. Furthermore, smart devices have made people more reliant on their mobile phones in the information age. The continuous updating of social platforms and entertainment methods has split time into unlimited fragments. Short videos on video streaming sites have a variety of forms, and they can be searched and are richer in content than other sources. Compared to clinical situations in which physicians provide patients with medical information orally or in writing and patients frequently cannot comprehend or retain the data, mobile short videos feature concise and clear images and may be viewed multiple times. They can therefore comprehensively satisfy the informational needs of all types of people. The literature indicates that a considerable proportion of patients search the internet for health information [19–21].

YouTube is the most popular video-sharing site with more than 2 billion views per day [22], and it is one of the sites most visited by people seeking medical information [23]. YouTube is open access, which means that user-generated content is not peer reviewed; while it offers effective health education resources, there may also be a great deal of misinformation and inaccuracy [22]. Consequently, numerous studies have evaluated the informational quality of YouTube videos with regard to oral-related topics [24–26]. To our knowledge, however, no recent study has evaluated content related to bleeding gums on YouTube™. Therefore, in this study, we aimed to conduct quantitative and qualitative analyses of YouTube videos related to bleeding gums.

## 2. Materials and methods

### 2.1. Research design

For this investigation, a cross-sectional design was used. Because it may be difficult for users to differentiate the quality of videos, we analyzed the videos' parameters and sources as well as their quality and reliability with the goal of conducting quantitative and qualitative analyses of YouTube videos related to bleeding gums.

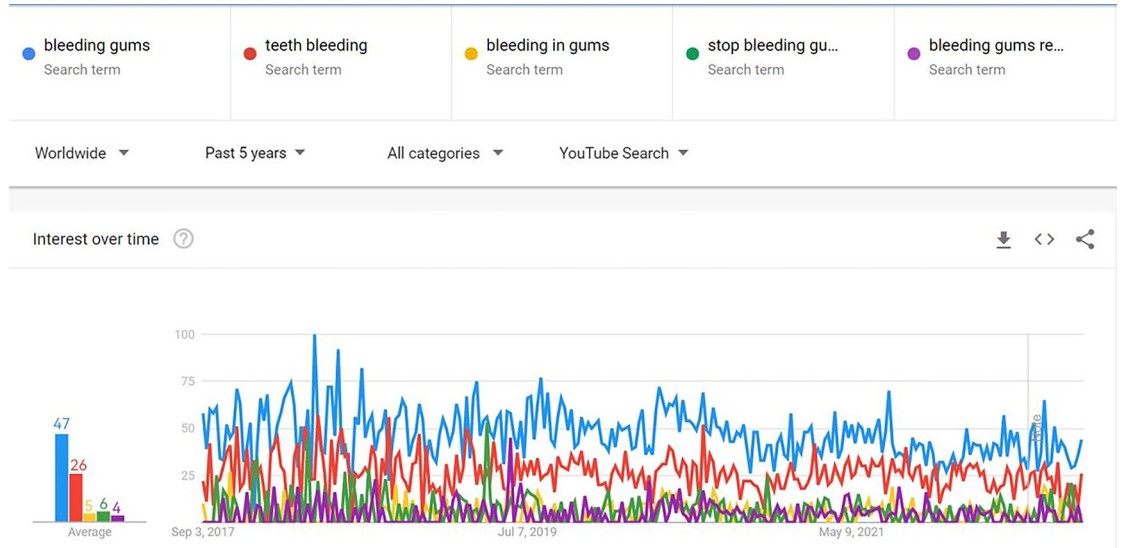

**Fig 1. Interest rates in different phrases over time in Google Trends.**

## 2.2. Search strategy

Google Trends (https://Trends.Google.com) is an online search engine designed to determine the frequency of searches for phrases over a certain period of time. Google Trends was used to identify the most popular preferred search keywords [27]. A YouTube search for "bleeding gums" was conducted on 28 August 2022 (one day). "Bleeding gums," "teeth bleeding," "bleeding in gums," "stop bleeding gums," and "bleeding gums remedy" were the most regularly used terms for this topic. According to Google Trends, when the search parameters were set for "past 5 years," "worldwide," "all categories," and "YouTube searches," "bleeding gums" was the most frequently used search term (Fig 1). Two researchers used the Microsoft Edge browser to browse "https://www.youtube.com." The computer history and cookies were deleted. YouTube searches were performed using the identified search terms. Filters are rarely used when non-experts search the internet [28, 29]. This study evaluated videos using YouTube's default "relevance" filter.

The results of the search query were displayed in order of site relevance, as evaluated by the YouTube website using a mixture of characteristics such as "views," "viewing rate," and "upload date." Commercial advertisements displayed on the side, above, or at the end of search results by YouTube were ignored and eliminated from the analysis. According to previous studies, the majority of YouTube viewers search for the top 60 to 200 videos [30]. Based on these data, the top 200 videos for the search terms were viewed and reviewed, and their links were kept for future reference. Notably, YouTube considers how closely the title, description, and video fit the query criteria and the query's engagement and watching duration, resulting in a dynamic ranking [31].

## 2.3. Inclusion and exclusion criteria (Fig 2)

A preliminary screening was performed to identify videos related to bleeding gums.

The inclusion criteria were as follows [32, 33]: 1. video quality of 240p or more; 2. videos in English (not in any other language, such as Chinese, Arabic, or French) that were translated using iFLYTEK-SR302 and passed manual calibration (the details, including web links, of the 107 videos on bleeding gums on YouTube are listed in the S1 Table; and 3. video duration

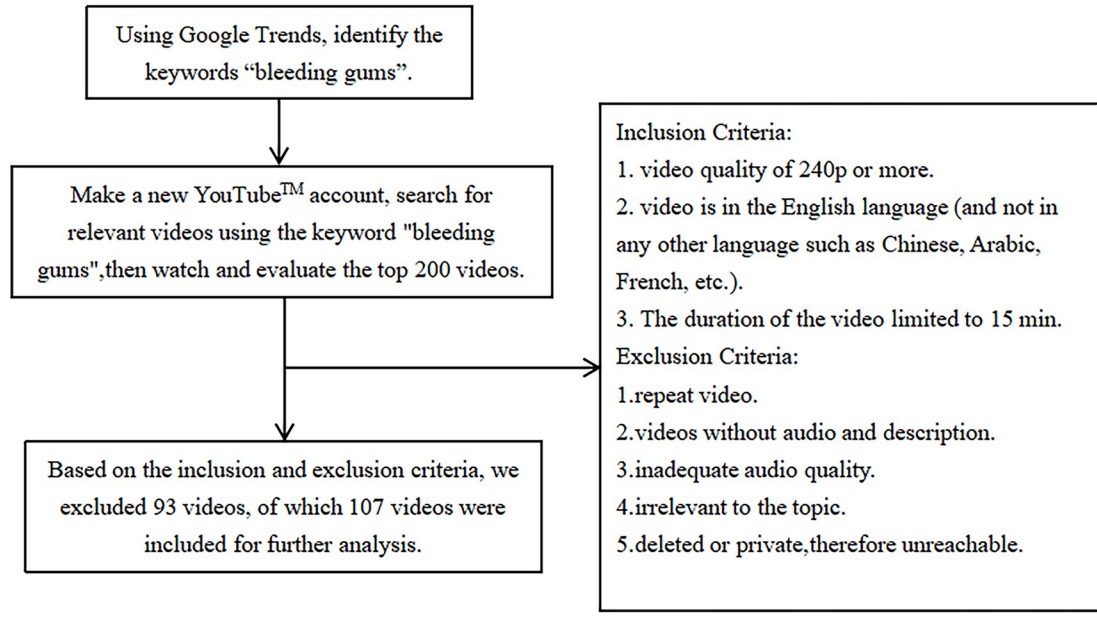

**Fig 2. A flow chart showing the screening process for YouTube™ videos.**

limited to 15 min (longer videos are less likely to capture the attention of online viewers according to a previous study) [26].

The following exclusion criteria were applied [34]: 1. repetition of videos; 2. videos without audio or descriptions; 3. inadequate audio quality; 4. irrelevant to the topic; and 5. deleted or private and therefore inaccessible.

## 2.4. Video parameters

The video upload date, length (in min), and number of likes, views, comments, subscribers, and viewing rate were retrieved. The viewing rate was determined using the techniques provided in prior research [22, 35]:

$$\frac{number\ of\ views}{number\ of\ days\ since\ upload} \times 100\%$$

## 2.5. Sources of the videos

The videos were classified by the upload source: hospital/clinic, dentist/specialist, professional organization/association/university, business (i.e., dental manufacturing company or dental supply company), health information website, TV channel or news agency, and others.

## 2.6. Assessment of quality

The Global Quality Score (GQS), a five-component scale used to estimate the educational value of each video, was utilized to assess the quality of all selected videos [36] (Table 1). Since the videos were primarily intended for nonprofessional audiences, the five main contents of the videos were evaluated systematically based on the usefulness score, which included local and systemic etiologies, symptoms, treatment, and prognosis. The content was scored as 0 (not

**Table 1. Global quality score (GQS) criteria.**

| GQS Description | GQS Score |
|---|---|
| Overall poor quality, poor flow of video, most of the information missing, not at all useful for patients | 1 |
| Generally poor quality and poor flow, some information listed but many important topics missing, of minimal use to patients | 2 |
| Moderate quality, suboptimal flow, some vital information adequately discussed but others poorly discussed, somewhat beneficial for patients | 3 |
| Good quality and generally good flow, most relevant information listed but some topics not covered, beneficial for patients | 4 |
| Excellent quality and flow, very useful for patients with complete information | 5 |

mentioned), 1 (brief introduction) or 2 (detailed introduction) depending whether each item was specifically discussed (Table 2). The combination of the global quality score (GQS) and the usefulness score determined the total score (Table 3). The total score was categorized as "poor" (0 to 5), "moderate" (6 to 10), or "good" (11 to 15 points). When the two researchers disagreed, discussions were held until consensus was reached.

## 2.7. Assessment of reliability

To assess the dependability of the YouTube videos, a modified version of the DISCERN tool developed by Charnock et al. [37] was used. The modified DISCERN reliability tool consists of five questions, each of which requires a yes or no response [37, 38]. A five-item questionnaire (score: 1 to 5) was used to evaluate health information. All "yes" responses were worth 1 point. The maximum possible score was 5 [37, 38] (Table 4). A score of 5 indicated excellent reliability, 4 indicated good reliability, 3 indicated moderate reliability, 2 indicated generally poor reliability, and a score of 1 indicated poor reliability [39].

## 2.8. Statistical analysis

IBM SPSS Statistics 27 software was used for statistical tests, and descriptive statistics were calculated for all the variables described above. The normality of the data was determined using the Shapiro–Wilk test. Non-parametric tests were used for data that did not adhere to the normal distribution: the Kruskal-Wallis test (for more than two samples), the Mann-Whitney test (for two independent samples), and Spearman's correlation coefficient. The Kruskal–Wallis test was used to compare video characteristics between "poor", "moderate" and "good" videos. The Mann–Whitney test was used to determine differences between videos' variable characteristics and length, and Spearman's correlation coefficient was used to assess correlations

**Table 2. Usefulness score.**

| Scoring item | Score |
|---|---|
| Local etiologies | 0/1/2 |
| Systemic etiologies | 0/1/2 |
| Symptoms | 0/1/2 |
| Treatment | 0/1/2 |
| Prognosis | 0/1/2 |
| Total | 0–10 |

(0, not mentioned; 1, briefly introduced; 2, introduced in detail)

**Table 3. Total score.**

| Total Score Description | Total Score |
|---|---|
| Poor | 0–5 |
| Moderate | 6–10 |
| Good | 11–15 |

between general YouTube video characteristics. Kappa coefficients of correlation were calculated to assess interrater reliability. The significance level was set at p <0.05.

## 2.9. Ethics statement

This research did not include humans or animals. The research examined YouTube videos that were open to the public. As a result, ethics committee approval was not necessary for this research. The data collection and analysis technique conformed with the rules and regulations of both Google Trends and YouTube.

## 3. Results

All videos were independently checked and evaluated by two researchers. The kappa coefficient used to measure interrater reliability was 0.674 for the video quality assessment.

Based on the inclusion and exclusion criteria, we excluded 93 (46.5%) videos (reasons for video exclusion are listed in the S2 Table). A total of 107 (53.5%) videos were included in further assessments. Among the 93 videos that were eliminated, 9.5% (19) were duplicates, 10% (20) lacked sound or explanation, 10% (20) had poor sound quality, 8.5% (17) were irrelevant to the topic, 6.5% (13) were not in English, 1.5% (3) were longer than 15 min, and one video was inaccessible because it had been deleted or because of privacy breaches (Table 5).

Descriptive statistics for the 107 YouTube videos that were assessed are shown in Table 6. The average number of days since upload of videos on YouTube related to bleeding gums was 1,172 days, the average video length was 3.04 min, the average number of likes was 624, the average number of views was 81,610, the average number of comments was 59, and 11 videos had closed comments. The average number of subscriptions was 355,696, the average viewing rate was 23,464.14%, the average DISCERN was 2.47, the average usefulness score was 4.53, the average GQS was 2.89, 75.70% of the videos (n = 81) had GQS scores less than 4 (Table 7), and the average total score was 7.42.

In light of the usefulness score, five specific components of the videos were systematically assessed. The highest coverage was for symptoms (95.33%), which were described in detail in 42 videos, followed by management (90.65%), local etiology (79.44%) and systemic etiology (38.32%). Prognosis was discussed in 16.82% of the videos but was not described in detail in all videos (Table 8).

In terms of the total score, only 14.02% of the videos received a "good" grade overall, followed by 55.14% "moderate" and 30.84% "poor." Most videos (n = 69, 64.48%) were uploaded

**Table 4. DISCERN reliability tool (1 point per question for yes answers).**

| |
|---|
| Are the explanations given in the video clear and understandable? |
| Are useful reference sources given (publication cited, from valid studies)? |
| Is the information in the video balanced and neutral? |
| Are additional sources of information given from which the viewer can benefit? |
| Does the video evaluate areas that are controversial or uncertain? |

**Table 5. Reasons for excluding videos for each search term.**

| Reasons for excluding | N | % |
|---|---|---|
| Repeat video | 19 | 9.5 |
| Videos without audio and description | 20 | 10 |
| Inadequate audio quality | 20 | 10 |
| Irrelevant to the topic | 17 | 8.5 |
| Deleted or private, therefore unreachable | 1 | 0.5 |
| Long video (>15 min) | 3 | 1.5 |
| Non-English | 13 | 6.5 |
| Total | 93 | 46.5 |

by hospitals/clinics and dentists/specialists. Among the videos uploaded by hospitals/clinics, 20 (68.97%) were classified as "moderate" and 3 (7.69%) as "good." Among the videos uploaded by dentists/specialists, 18 (45.00%) were classified as "moderate" and 8 (20.00%) as "good"; 1 video uploaded by an association was rated as "poor" due to its short duration, and the content mentioned was relatively limited. Eight (61.54%) of the videos uploaded by commercials were in the "poor" category (Table 9).

The Kruskal–Wallis test was used to compare the characteristics of the "poor," "medium," and "good" videos, as shown in Table 10. Videos rated "good" had a shorter mean time (866.13 min) after upload, videos rated "medium" had a longer mean time after upload (1,255.15 min), and there was no significant difference between the groups (p >0.05). Similarly, the average viewing rate of videos rated "poor" (63,943.68%) was significantly greater than that of the other groups (p <0.05). The average length of the videos, the average number of likes, and the average number of comments on the "poor," "medium" and "good" videos were significantly different (p <0.05).

As shown in Table 11, video length was positively correlated with the number of likes, viewership, usefulness score, GQS, total score, and DISCERN (p <0.05). Video length was moderately correlated with the usefulness score (r = 0.605), GQS (r = 0.554), total score (r = 0.617) and DISCERN (r = 0.424). The number of likes was strongly correlated with the viewing rate (r = 0.920, p <0.05). The total score was strongly correlated with the usefulness score and GQS (r = 0.980, p <0.05; r = 0.923, p <0.05) and moderately correlated with DISCERN (r = 0.610, p <0.05). As shown in Table 12, the viewing rate for videos greater than 4 min in length was extremely weakly correlated with video length (r = 0.044), and the viewing rate for videos less

**Table 6. Descriptive variables of the YouTube™ videos.**

| | Minimum | Maximum | Mean | SD |
|---|---|---|---|---|
| Days since upload | 31 | 4731 | 1172.45 | 915.55 |
| Video length (min) | 0.06 | 12.2 | 3.04 | 2.69 |
| No. of likes | 0 | 23000 | 624.07 | 2432.33 |
| No. of views | 3 | 4476045 | 81610.81 | 462670.97 |
| No. of comments | 0 | 2030 | 59.49 | 227.05 |
| No. of subscriptions | 2 | 8500000 | 355696.78 | 1102613.67 |
| Viewing rate (%) | 1.77 | 2081881.40 | 23464.14 | 201237.61 |
| Usefulness score | 1 | 9 | 4.53 | 1.98 |
| GQS | 1 | 5 | 2.89 | 0.93 |
| Total score | 2 | 14 | 7.42 | 2.79 |
| DISCERN | 0 | 4 | 2.47 | 0.89 |

**Table 7. Frequency and percentage of the global quality scale score.**

| Item Scored | n (%) |
|---|---|
| 1. Overall poor quality, poor flow of video, most information missing, not at all useful for patients | 7 (6.54%) |
| 2. Generally poor quality and poor flow, some information listed but many important topics missing, of minimal use to patients | 32 (29.91%) |
| 3. Moderate quality, suboptimal flow, some vital information adequately discussed but other information poorly discussed, somewhat beneficial for patients | 42 (39.25%) |
| 4. Good quality and generally good flow, most relevant information listed but some topics not covered, beneficial for patients | 22 (20.56%) |
| 5. Excellent quality and flow, very useful for patients with complete information | 4 (3.74%) |

than 4 min in length was weakly correlated with video length (r = 0.275). However, neither was significantly different (p >0.05).

Table 13 highlights seven variables that differed significantly (p <0.05) between videos less than 4 min in length and those greater than 4 min in length, namely, the average number of likes, the average number of comments, the average number of views, the average DISCERN score, the average usefulness score, the average GQS, and the average total score. The data show that videos greater than 4 min in length had a greater average number of likes (1,807.53) and comments (170.30) and a lower average number of views (10,574.02). Videos longer than 4 min in length had a higher average DISCERN score, usefulness score, GQS, and overall score.

## 4. Discussion

To the best of our knowledge, this is the first study to explore the content of YouTube videos about bleeding gums. People commonly use the YouTube platform to search for health-related concerns due to the increased popularity of social networks and the internet and the ease and

**Table 8. Distribution of the usefulness score items.**

| | 0 point (Not mentioned) | 1 point (Briefly introduced) | 2 points (Introduced in detail) |
|---|---|---|---|
| Local etiologies | 22 | 52 | 33 |
| Systemic etiologies | 66 | 19 | 22 |
| Symptoms | 5 | 60 | 42 |
| Treatment | 10 | 52 | 45 |
| Prognosis | 89 | 18 | 0 |

**Table 9. Comparison of YouTube™ video uploaders between poor, moderate, and good informational videos about bleeding gums.**

| | Poor | Moderate | Good | Total (N/%) |
|---|---|---|---|---|
| Hospital/clinic | 6 (20.69%) | 20 (68.97%) | 3 (7.69%) | 29(27.10%) |
| Dentist/specialist | 14 (35.00%) | 18 (45.00%) | 8 (20.00%) | 40(37.38%) |
| Professional organization/association/university | 1 (100.00%) | / | / | 1(0.93%) |
| business | 8 (61.54%) | 4 (30.77%) | 1 (7.69%) | 13(12.15%) |
| Health information website | 1 (6.67%) | 11 (73.33%) | 3 (20%) | 15(14.02%) |
| TV channel or news agency | / | 2 (100.00%) | / | 2(1.87%) |
| Others | 3 (42.86%) | 4 (57.14%) | / | 7(6.54%) |
| Total | 33 (30.84%) | 59 (55.14%) | 15 (14.02%) | 107(100%) |

**Table 10. Comparison of video parameters among poor, moderate, and good informational videos about bleeding gums.**

| Parameters | Poor (n = 33) | | Moderate (n = 59) | | Good (n = 15) | | P |
|---|---|---|---|---|---|---|---|
| | Mean | SD | Mean | SD | Mean | SD | |
| Days since upload | 1163.82 | 901.27 | 1255.15 | 999.45 | 866.13 | 471.33 | 0.534 |
| Video length (min) | 1.38 | 1.49 | 3.18 | 2.41 | 6.17 | 2.97 | <0.001 |
| No. of likes | 137.58 | 390.41 | 438.41 | 914.85 | 2424.67 | 6066.45 | 0.011 |
| No. of comments | 11.11 | 24.43 | 36.93 | 73.07 | 243.29 | 556.95 | 0.002 |
| Viewing rate (%) | 63943.68 | 362261.74 | 3608.90 | 6817.96 | 12506.42 | 28451.11 | 0.023 |
| DISCERN | 1.73 | 0.98 | 2.76 | 0.63 | 2.93 | 0.594 | <0.001 |

The Kruskal–Wallis test (p value <0.05).

accessibility of the available information. The tendency to use such applications appears to have increased during the COVID-19 pandemic [39]. The risk of infection with COVID-19 is high in dental practice due to the close contact between dentists and patients and the formation of aerosols in dental practices [40–42]. This has led people to avoid dental clinics during the COVID-19 epidemic [40]. However, studies have found that periodontal diseases are associated with severe COVID-19-related complications [43]. Therefore, the American Academy of Periodontology (AAP) emphasizes the importance of maintaining periodontal health during the COVID-19 outbreak. Currently, YouTube shows an advantage in sharing medical content related to bleeding gums.

Moreover, since the cost of internet communication has declined significantly, there are objective reasons for people to consult online. In this study, the total number of views of the videos exceeded 8,732,400, the total number of likes was 66,800, and the videos received a total of 5,711 comments. These statistics indicate that most people learn about bleeding gums on YouTube. All registered users on YouTube are permitted to submit and distribute health-related videos for free without peer review. Inaccurate videos with no scientific basis are often uploaded on YouTube because the sources of the material are sometimes ambiguous [22].

**Table 11. Correlation of the data.**

| | | Video length (min) | No. of likes | Viewing rate (%) | Usefulness Score | GQS | Total Score |
|---|---|---|---|---|---|---|---|
| Video length (min) | r | | | | | | |
| | p | | | | | | |
| No. of likes | r | 0.294 | | | | | |
| | p | 0.002 | | | | | |
| Viewing rate (%) | r | 0.229 | 0.920 | | | | |
| | p | 0.017 | <0.001 | | | | |
| Usefulness Score | r | 0.605 | 0.220 | 0.144 | | | |
| | p | <0.001 | 0.023 | 0.139 | | | |
| GQS | r | O.554 | 0.577 | 0.550 | 0.837 | | |
| | p | <0.001 | <0.001 | <0.001 | <0.001 | | |
| Total Score | r | 0.617 | 0.355 | 0.286 | 0.980 | 0.923 | |
| | p | <0.001 | <0.001 | 0.003 | <0.001 | <0.001 | |
| DISCERN | r | 0.424 | 0.179 | 0.089 | 0.580 | 0.566 | 0.610 |
| | p | <0.001 | 0.066 | 0.359 | <0.001 | <0.001 | <0.001 |

r: Spearman's rank correlation coefficient (p value <0.05).

**Table 12. Comparison between video length and viewing rate of YouTube™ videos on bleeding gums.**

| | | | Video length (min) | Viewing rate (%) |
|---|---|---|---|---|
| <4 min (n = 77) | Video length (min) | r | 1 | 0.044 |
| | | p | - | 0.703 |
| | Viewing rate (%) | r | 0.044 | 1 |
| | | p | 0.703 | - |
| >4 min (n = 30) | Video length(min) | r | 1 | 0.275 |
| | | p | - | 0.142 |
| | Viewing rate (%) | r | 0.275 | 1 |
| | | p | 0.142 | - |

r: Spearman's rank correlation coefficient (p value<0.05).

Thus, this study quantitatively and qualitatively analyzed YouTube videos about bleeding gums.

This study combined the GQS with the usefulness score to assess the included videos and used the DISCERN score to rate the videos' reliability. Similar to earlier studies, GQS and DISCERN were used in recent studies [25, 44, 45]. In terms of the analysis of upload sources, the majority (n = 69, 64.48%) of the observed videos were uploaded by hospitals/clinics and dentists/specialists, indicating that with the development of video sites, medical professionals are aware of the importance of these sites for the dissemination of expertise. More than 60% of the videos uploaded by hospitals/clinics and dentists/specialists were moderate or above, while the percentage of videos uploaded by businesses or others was "poor," indicating that the videos uploaded by dental clinicians are higher quality and more comprehensive. Previous studies have shown that high-quality videos are created by medical experts or organizations, while low-quality videos are created by medical advertising, for-profit organizations, and individual users [44]. These findings are consistent with our study. The mean DISCERN value for the videos was 2.47. The low scores reflect the low reliability of videos on YouTube about bleeding gums. The DISCERN score was positively connected with the total video score (p <0.05), indicating that the reliability of the videos was positively connected with the video quality. To evaluate the overall quality of the video, we used the GQS and the usefulness score. The average GQS was 2.89 and the average total score was 7.42, indicating moderate quality. Various outcomes have been observed in prior research. In research conducted by Ramadhani et al. [46], lay users posted the majority of videos about bad breath which can result from oral or extra-oral sources [47]. The study results showed poor quality, poor reliability, and inadequate video

**Table 13. Comparison of video variables based on duration.**

| | <4 min (n = 77) | | >4 min (n = 30) | | P |
|---|---|---|---|---|---|
| | Mean | SD | Mean | SD | |
| No. of likes | 162.99 | 361.46 | 1807.53 | 4389.66 | 0.013 |
| No. of comments | 16.13 | 39.17 | 170.30 | 408.19 | <0.001 |
| Viewing rate (%) | 28486.26 | 237112.34 | 10574.02 | 21024.21 | 0.017 |
| Usefulness Score | 4.03 | 1.77 | 5.83 | 1.91 | <0.001 |
| GQS | 2.65 | 0.86 | 3.50 | 0.84 | <0.001 |
| Total score | 6.68 | 2.51 | 9.33 | 2.60 | <0.001 |
| DISCERN | 2.32 | 0.92 | 2.83 | 0.70 | 0.014 |

Mann–Whitney test (p value<0.05).

content. This may be because lay users have limited medical knowledge, so the quality of the uploaded videos is poor. In studies conducted by Yavuz et al. [48] and Kurian et al. [49], dental professionals mainly uploaded the videos. The quality of the videos on accelerated orthodontic treatment in the study by Yavuz et al. [48] was "good," probably because the videos uploaded by dental professionals were mainly educational and the study did not exclude videos of long duration. However, the results of Kurian et al. [49] showed that videos on fixed implant-supported prostheses were poor quality because dental professionals uploaded these videos with a marketing bias rather than providing purely educational information. Most of the videos in our analysis were also posted by dental clinicians. Fourteen of the videos uploaded by dental clinicians classified as "good" contained comprehensive, high-quality, and mainly educational content. However, most of the videos on bleeding gums were not comprehensive. This may be related to the exclusion of videos >15 min, which may contain more comprehensive content. Some of the evaluated videos were uploaded by nonprofessional users, which also affected the overall quality of the evaluated videos. Videos uploaded by businesses were mainly marketing, and videos uploaded by others were of poor quality due to their limited medical knowledge.

The content evaluation process revealed only a few videos that described the systemic etiology and prognosis, none of which provided a detailed prognosis. Only six of the videos contained all the content, indicating that the content of YouTube videos on bleeding gums is not comprehensive. A thorough understanding of the systemic etiology and prognosis is essential to treat bleeding gums. Bleeding gums can be caused by local and systemic factors. Therefore, it is essential to pay attention not only to the local etiology but also to understand the patient's systemic situation, clarify the etiology, and treat it promptly to avoid misdiagnosis. Depending on the individual and the degree of periodontal disease, the periodontal prognosis involves making a "forecast" of periodontal disease. The maintenance programs and treatment designs used differ. Increased intervention efforts and identification of high-risk teeth or individuals promote disease improvement.

In the present study, the average length of videos rated as "good" was significantly greater than that of the other groups. Video length was moderately correlated with GQS, the usefulness score, the total score, and the DISCERN score. This may be because the longer a video is, the more content can be explained and the higher the quality of the video. The length of the video not only has an impact on the quality of the video but also plays a significant role in the viewer's attention. On the YouTube platform, a user-selectable filter is available for video length, i.e., videos less than 4 min and greater than 4 min (if the video is less than 4 min, YouTube will mark it as a short video). Therefore, in this study, we divided the videos into two groups of less than 4 min and more than 4 min for analysis. Videos longer than 4 min had a higher average DISCERN, usefulness score, GQS, and total score but a lower average number of views, indicating that longer videos may reduce viewers' attention levels, while videos longer than 4 min have better quality and reliability. However, videos less than 4 min in length tend to be watched heavily and have poorer video quality and reliability. A previous study by Ajumobi et al. [50] showed that videos under 4 min were more succinct and captured the attention of a broader audience. In comparison, videos longer than 4 min were of better quality, consistent with the results of our study. In a previous study by Delli et al. [51], the duration of videos that were found to be beneficial to viewers was approximately 7 min. For this reason, it is crucial to consider both the content and the duration when producing videos so that they are educational and engaging for the audience.

Incorrect and inadequate content in YouTube videos about bleeding gums can significantly influence patients' attitudes and medical decisions. Therefore, it is imperative to monitor health information on YouTube. To avoid inaccurate health-related material on social media platforms, it is advised that relevant medical experts and government organizations monitor

social networks or use other valid methods to evaluate video content. However, no such implementation has been developed to date [52, 53]. Therefore, we believe that the video quality of the YouTube platform can be improved by involving more dental professionals and organizations in producing high-quality videos related to dental problems. Furthermore, YouTube can offer a certification feature that assesses the quality of videos uploaded by dentists and professional institutions. This evaluation will be conducted using advanced Deep Learning models [54], which rely on sophisticated architecture and a meticulously organized YouTube dataset. Videos that meet the certification standards can be awarded an official mark, ensuring users that they are reliable and trustworthy resources for information on dental-related topics. When users search for bleeding gum-related problems, officially certified videos will be highlighted. Additionally, dentists may recommend videos of higher quality and reliability to their patients based on information they find on YouTube. YouTube should continue to serve as a reliable source of dental information.

The current study has some limitations. First, YouTube search results change as recent videos are added or removed. Therefore, new video additions or modifications may have occurred after the date the included videos were analyzed, resulting in slight differences. Second, patients may type different search phrases and receive different results. To minimize this limitation, we used Google Trends to select our audience's most frequently searched terms. Third, YouTube recently removed the ability to view the number of video dislikes. Therefore, dislike data and interaction indices cannot be displayed. Fourth, we excluded several videos that were >15 min in length, which might have offered excellent content. Finally, the videos we filtered with our translation software were somewhat skewed in scope but covered most of the YouTube videos.

## Supporting information

**S1 Table.**
(XLSX)

**S2 Table.**
(XLSX)

## Acknowledgments

We thank Dr. Weijun Zheng (Department of Medical Statistics, Zhejiang Chinese Medicine University, China) for providing professional instructions in statistical analyses.

## Author Contributions

**Conceptualization:** Jiali Wu, Danlin Li, Minkui Lin.

**Data curation:** Jiali Wu, Danlin Li.

**Formal analysis:** Jiali Wu, Danlin Li.

**Investigation:** Jiali Wu, Danlin Li, Minkui Lin.

**Methodology:** Minkui Lin.

**Project administration:** Jiali Wu, Danlin Li, Minkui Lin.

**Writing – original draft:** Jiali Wu, Danlin Li.

**Writing – review & editing:** Jiali Wu, Danlin Li, Minkui Lin.

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
