## [Decision Letter · Decision Letter 0]

1 Aug 2023

PONE-D-23-16369YouTube as an information source for bleeding gums: a quantitative and qualitative analysisPLOS ONE

Dear Dr. Lin,

Thank you for submitting your manuscript to PLOS ONE. After careful consideration, we feel that it has merit but does not fully meet PLOS ONE’s publication criteria as it currently stands. Therefore, we invite you to submit a revised version of the manuscript that addresses the points raised during the review process.

We look forward to receiving your revised manuscript.

Kind regards,

Anand Marya, BDS, MScD Orthodontics

Academic Editor

PLOS ONE

2. In your Methods section, please include additional information about your dataset and ensure that you have included a statement specifying whether the collection and analysis method complied with the terms and conditions for the source of the data.

Additional Editor Comments:

Dear Authors,

Please address the comments of the reviewers and offer a detailed point by point response to their comments.

Best Wishes

Reviewers' comments:

Reviewer's Responses to Questions

**Comments to the Author**

1. Is the manuscript technically sound, and do the data support the conclusions?

Reviewer #1: No

Reviewer #2: Yes

2. Has the statistical analysis been performed appropriately and rigorously? 

Reviewer #1: I Don't Know

Reviewer #2: Yes

3. Have the authors made all data underlying the findings in their manuscript fully available?

Reviewer #1: Yes

Reviewer #2: Yes

4. Is the manuscript presented in an intelligible fashion and written in standard English?

Reviewer #1: No

Reviewer #2: Yes

5. Review Comments to the Author

Reviewer #1: Dear Author,

What is novelty of your research ?? the methodology seems vague particularly study design?? Clarify it. the inclusion and exclusion criteria also needs to be clarified. How you accessed the video quality?? is there any gold standrad??

Reviewer #2: This is an interesting study where the researchers conducted a quantitative and qualitative analysis of YouTube videos related to bleeding gums.

Introduction

“According to the new classification of periodontal and peri-implant diseases and conditions for 2017.” Please find the latest classification.

Gingivitis, Periodontitis and peri-implantitis are common diseases. Peri-implantitis also can result in bleeding gums. Please add this.

https://pubmed.ncbi.nlm.nih.gov/32882741/

https://www.ncbi.nlm.nih.gov/pmc/articles/PMC7587339/

Method

“A YouTube search for "bleeding gums" was conducted on 28 August 2022”. Is the search done only one day or for a specific duration? Please elaborate.

“The duration of the video was limited to 15 min (longer videos were less likely to capture the attention of online viewers, according to a previous study, so the video was restricted to 15 min).” This is not really true. Some patient having bleeding gum if wants to know more may want to know more about it can see or listen the longer and detail videos.

How many researchers did the search. Needs to add.

6. PLOS authors have the option to publish the peer review history of their article (what does this mean?). If published, this will include your full peer review and any attached files.

Reviewer #1: No

Reviewer #2: No

---

## [Author Response · Author response to Decision Letter 0]

14 Sep 2023

Dear Editor:

We would like to express our sincere gratitude to Dr. Anand Marya for accepting the role of editor for our submitted manuscript titled "YouTube as an information source for bleeding gums: a quantitative and qualitative analysis" (PLOS ONE manuscript number: PONE-D-23-16369). Furthermore, we truly appreciate that you had assigned such qualified reviewers to our manuscript, whose insightful comments and constructive suggestions greatly assisted us in the revision process. We extend our heartfelt thanks for their invaluable contribution. Our point-by-point responses to the comments are provided as a separate file labeled 'Response to Reviewers'.

Thank you for your consideration. I look forward to hearing from you.

Best Regards.

Prof. Minkui Lin

Email address: linmk105@sina.com.

---

## [Decision Letter · Decision Letter 1]

21 Nov 2023

PONE-D-23-16369R1YouTube as an information source for bleeding gums: a quantitative and qualitative analysisPLOS ONE

Dear Dr. Lin,

Thank you for submitting your manuscript to PLOS ONE. After careful consideration, we feel that it has merit but does not fully meet PLOS ONE’s publication criteria as it currently stands. Therefore, we invite you to submit a revised version of the manuscript that addresses the points raised during the review process.

**I congratulate you on submitting the revised manuscript. However, there are still glaring deficiencies which need to be addressed. I am hoping that you address those.**

We look forward to receiving your revised manuscript.

Kind regards,

Tanay Chaubal

Academic Editor

PLOS ONE

Reviewers' comments:

Reviewer's Responses to Questions

**Comments to the Author**

1. If the authors have adequately addressed your comments raised in a previous round of review and you feel that this manuscript is now acceptable for publication, you may indicate that here to bypass the “Comments to the Author” section, enter your conflict of interest statement in the “Confidential to Editor” section, and submit your "Accept" recommendation.

Reviewer #1: (No Response)

Reviewer #3: (No Response)

2. Is the manuscript technically sound, and do the data support the conclusions?

Reviewer #1: Partly

Reviewer #3: Yes

3. Has the statistical analysis been performed appropriately and rigorously? 

Reviewer #1: I Don't Know

Reviewer #3: Yes

4. Have the authors made all data underlying the findings in their manuscript fully available?

Reviewer #1: No

Reviewer #3: Yes

5. Is the manuscript presented in an intelligible fashion and written in standard English?

Reviewer #1: No

Reviewer #3: Yes

6. Review Comments to the Author

Reviewer #1: Dear Author,

The rebuttal against the raised queries are not satisfactory and not addressed as expected.

Reviewer #3: introduction line 29- its not explanatory, patients frequently present to the dental clinic with bleeding gums while brushing their teeth or noticing blood in their mouth in the morning after waking up.

line 33- classification is 2017 not 2018

what were your keywords for search?

7. PLOS authors have the option to publish the peer review history of their article (what does this mean?). If published, this will include your full peer review and any attached files.

Reviewer #1: No

Reviewer #3: No

---

## [Author Response · Author response to Decision Letter 1]

5 Jan 2024

Dear Editor:

Thank you for giving us the opportunity to submit a revised version of “YouTube as a Source of Information on Bleeding Gums: A Quantitative and Qualitative Analysis” (PLOS ONE manuscript number: PONE-D-23-16369) for potential publication in PLOS ONE. We appreciate your and the reviewers’ time and effort in providing feedback on our article. Your views, as well as the reviewers’, are valuable and have been instrumental in improving our research papers. We have carefully reviewed all of the suggestions and have made our best efforts to incorporate them into the text in order to meet the acceptance criteria. The following is our peer-to-peer response to these comments (Detailed response is provided as a separate document labeled "Responses to Reviewers." ):

Reviewer Comments and Responses:

1.If the authors have adequately addressed your comments raised in a previous round of review and you feel that this manuscript is now acceptable for publication, you may indicate that here to bypass the “Comments to the Author” section, enter your conflict of interest statement in the “Confidential to Editor” section, and submit your "Accept" recommendation.

Reviewer #1: (No Response)

Reviewer #3: (No Response)

Our response: None .

2.Is the manuscript technically sound, and do the data support the conclusions? 

Reviewer #1: Partly

Reviewer #3: Yes

Our response: We appreciate the reviewer’s acknowledgement of our efforts to improve the manuscript based on the comments. However, there are still some areas that require clarification. 

Firstly, we would like to point out that our choice of viewing and reviewing the top 200 videos was based on a study [1] indicating that the majority of YouTube viewers search within this range. This allowed us to have a suitable sample size for analysis. 

Furthermore, the analytical techniques employed in our study are well-established. For instance, we utilized the global quality score (GQS) [2,3], usefulness score [4], and DISCERN [5], all of which have been extensively used in peer-reviewed studies and serve to strengthen and enhance our conclusions. 

Finally, we agree with the reviewer that collecting high-quality data is a critical initial step in establishing the validity of the findings. To ensure this, we had two researchers independently examine and rate each video, and the reliability of the raters was assessed before commencing the analysis of the videos. In light of the reviewer’s recommendation regarding the manuscript data, we have re-evaluated our statistical methods and experimental data.

To sum up, we tried our best to study the topic using classical methods from numerous literatures, and have obtained the data and information to reach the conclusions of this manuscript. 

References

1.Desai T, Shariff A, Dhingra V, et al. Is content really king? An objective analysis of the public's response to medical videos on YouTube. PLoS One. 2013; 8(12): e82469.

2.Sezici YL, Gediz M, Dindaroğlu F. Is YouTube an adequate patient resource about orthodontic retention? A cross-sectional analysis of content and quality. Am J Orthod Dentofacial Orthop. 2022; 161(1): e72-e79.

3.Ustdal G, Guney AU. YouTube as a source of information about orthodontic clear aligners. Angle Orthod. 2020; 90(3): 419-424. 

4.Çapan BŞ. YouTube as a source of information on space maintainers for parents and patients. PLoS One. 2021; 16(2): e0246431.

5.Charnock D, Shepperd S, Needham G, et al. DISCERN: an instrument for judging the quality of written consumer health information on treatment choices. J Epidemiol Community Health. 1999; 53(2): 105-111.

Revised text: not applicable. (The statistical methods and the revised text related are seen in our third response)

3.Has the statistical analysis been performed appropriately and rigorously?

Reviewer #1: I Don't Know

Reviewer #3: Yes

Our response: We sincerely apologize for not providing sufficient clarity regarding the appropriateness of our statistical analysis in our previous response letter. Prior to evaluating the videos, we consulted with statistical experts and carefully reviewed pertinent literature [1-4]. To ensure reliability among raters, we calculated Kappa coefficients of correlation. The Shapiro-Wilk test was employed to determine whether our data conformed to a normal distribution. For datasets that deviated from normality, we utilized non-parametric tests: specifically, the Kruskal-Wallis test for comparing more than two samples, the Mann-Whitney test for comparing two independent samples, and Spearman's correlation coefficient to assess relationships between variables.

In our revised manuscript, we have included extensive details on our statistical methodologies, elucidating the specific tests employed and the rationales behind their selection. Furthermore, to enhance transparency and readability, we have clearly indicated which statistical tests were used within each chart legend. This added clarity aims to facilitate a thorough understanding of our analytical approach and the robustness of our findings.

References

1.Lena Y, Dindaroğlu F. Lingual orthodontic treatment: A YouTube™ video analysis. Angle Orthod. 2018; 88(2): 208-214.

2.Kaval ME, Demirci GK, Atesci AA, et al. YouTube™ as an information source for regenerative endodontic treatment procedures: Quality and content analysis. Int J Med Inform. 2022; 161: 104732.

3.Sezici YL, Gediz M, Dindaroğlu F. Is YouTube an adequate patient resource about orthodontic retention? A cross-sectional analysis of content and quality. Am J Orthod Dentofacial Orthop. 2022; 161(1): e72-e79. 

4.Fortuna G, Schiavo JH, Aria M, et al. The usefulness of YouTube™ videos as a source of information on burning mouth syndrome.J Oral Rehabil. 2019; 46(7): 657-665.

4.Have the authors made all data underlying the findings in their manuscript fully available?

Reviewer #1: No

Reviewer #3: Yes

Our response: We are immensely grateful for your thoughtful reminder, as we recognize that due to space constraints, it is not always possible to include all relevant information in the main manuscript. We strive to provide any previously omitted information wherever feasible. Unfortunately, the size of the 200 videos in this study, amounting to 2.78GB, exceeded the file upload limit of PLOS ONE, preventing us from uploading them. Therefore, we have included comprehensive details of the 107 included videos in the S1 Table and provided reasons for the exclusion of 93 videos in the S2 Table.

5.Is the manuscript presented in an intelligible fashion and written in standard English?

Reviewer #1: No

Reviewer #3: Yes

Our response: Thank you for your thoughtful suggestion. Based on your recommendation, we have re-engaged a professional editorial team (AJE editorial team) to assist us in polishing the manuscript. We have thoroughly revised the entire manuscript to ensure that it is easy to understand and written in standard English. We sincerely hope that the revised manuscript will meet your satisfaction.

6.Review Comments to the Author

Reviewer #1: Dear Author,

The rebuttal against the raised queries are not satisfactory and not addressed as expected.

We have carefully reconsidered the 1st round queries from Reviewer #1 and aim to ensure the following updated responses fully satisfies your needs. 

(1) What is novelty of your research ? 

Our response: We appreciate the opportunity to explain the novelty of our research and to make revisions. We noted that our previous abstract did not highlight the novelty of this paper, so we revised the abstract to highlight what is novel about our study.

In short, bleeding gums present a prevalent dental issue in individuals' everyday lives [1-3]. The heightened risk of COVID-19 infection arises from the close proximity between dental practitioners and patients, as well as the generation of aerosols within dental settings [4-6]. Consequently, individuals have refrained from visiting dental clinics amidst the COVID-19 pandemic [4]. In this context, the utilization of the internet can effectively mitigate transmission rates by minimizing physical contact, The utilization of the YouTube platform for the purpose of seeking health-related information has become prevalent among individuals, and it is important to note that YouTube operates on an open-access model, wherein user-generated content lacks the scrutiny of peer review. While YouTube effectively provides valuable health education resources, it is imperative to acknowledge the potential prevalence of misinformation and inaccuracies [7]. Dental professionals have acknowledged the significance of YouTube as a source of patient information, consequently, numerous studies have evaluated the informational quality of YouTube videos on oral-related topics such as burning mouth syndrome, root canal treatment, lingual orthodontic treatment, regenerative endodontic treatment et al [8-11]. According to our knowledge, no recent study has looked into bleeding gums-related content on YouTube. Incorrect and inadequate content in YouTube videos about bleeding gums can significantly influence patients' attitudes and medical decisions. Consequently, the objective of this study is to perform both quantitative and qualitative analyses of YouTube videos pertaining to bleeding gums.

In our revision, we have directly stated the novelty of our research in the abstract (lines 11-13), introduction (lines 61-64), and discussion (lines 248-249).

References

1.Lawal FB, Dosumu EB. Self-reported and clinically evident gingival bleeding and impact on oral health-related quality of life in young adolescents: a comparative study. Malawi Med J. 2021; 33(2): 121-126.

2.Molina GF, Faulks D, Mazzola I, et al. Three-year survival of ART high-viscosity glass-ionomer and resin composite restorations in people with disability. Clin Oral Investig. 2018; 22(1): 461-467.

3.Chen H, Zhang R, Cheng R, et al. Gingival bleeding and calculus among 12-year-old Chinese adolescents: a multilevel analysis. BMC Oral Health. 2020; 20(1): 147.

4.Loch C, Kuan IB, Elsalem L, et al. COVID‐19 and dental clinical practice: Students and clinical staff perceptions of health risks and educational impact. J Dent Educ. 2021; 85(1): 44-52.

5.Haridy R, Abdalla MA, Kaisarly D, et al. A cross‐sectional multicenter survey on the future of dental education in the era of COVID‐19: Alternatives and implications. J Dent Educ. 2021; 85(4): 483-493.

6.Iyer P, Aziz K, Ojcius DM. Impact of COVID‐19 on dental education in the United States. J Dent Educ. 2020; 84(6): 718-722.

7.Hassona Y, Taimeh D, Marahleh A, et al. YouTube as a source of information on mouth (oral) cancer. Oral Dis. 2016; 22(3): 202-208.

8.Fortuna G, Schiavo JH, Aria M, et al. The usefulness of YouTube™ videos as a source of information on burning mouth syndrome. J Oral Rehabil. 2019; 46(7): 657-665.

9.Nason K, Donnelly A, Duncan HF. YouTube as a patient-information source for root canal treatment. Int Endod J. 2016; 49(12): 1194-1200.

10.Lena Y, Dindaroğlu F. Lingual orthodontic treatment: A YouTube™ video analysis. Angle Orthod. 2018; 88(2): 208-214.

11.Kaval ME, Demirci GK, Atesci AA, et al. YouTube™ as an information source for regenerative endodontic treatment procedures: Quality and content analysis. Int J Med Inform. 2022; 161: 104732.

(2) Concern of the reviewer: the methodology seems vague particularly study design? Clarify it. 

Our response: We are sorry that our research method, especially the study design, was not well explained in the previous response letter. Our study was aimed at a cross-sectional study of YouTube videos related to bleeding gums, as the initial portrayal of the study design section may potentially perplex readers, in light of this, we have made appropriate revisions (lines 68-71), referring to the previous study conducted by Antonio et al [1-3]. Our research design is as follows (The detailed research methods are included in the Materials and Methods section of our manuscript) :

1. Conducted a search on YouTube using the keyword "bleeding gums" from Google Trends, and collected samples using inclusion and exclusion criteria;

2. The descriptive statistics for the videos included the time since upload, the video length, and the number of likes, views, comments, subscribers, and viewing rates;

3. The global quality score (GQS), usefulness score, and DISCERN were used to evaluate the video quality;

4. Statistical analysis was performed using the Kruskal–Wallis test, Mann–Whitney test, and Spearman correlation analysis.

References

1.Romano A, Lauritano D, Fiori F, et al. Cross-sectional study on the quality of oral lichen planus videos on YouTube™. J Oral Pathol Med. 2021; 50(2): 220-228.

2.Sezici YL, Gediz M, Dindaroğlu F. Is YouTube an adequate patient resource about orthodontic retention? A cross-sectional analysis of content and quality. Am J Orthod Dentofacial Orthop. 2022; 161(1): e72-e79. 

3.Uzel İ, Ghabchi B, Akalın A, et al. YouTube as an information source in paediatric dentistry education: Reliability and quality analysis. PLoS One. 2023; 18(3): e0283300.

(3). Concern of the reviewer: the inclusion and exclusion criteria also needs to be clarified. 

Our response: We are sorry that our previous response letter did not clearly explain the inclusion and exclusion criteria of our study, and the updated explanation is as follows:

1. Because the results may be heavily influenced by inclusion and exclusion criteria, therefore, our study referred to research on information quality analysis of YouTube videos on oral related topics [1-4], which will undoubtedly make the conclusions more solid and reliable. More details on inclusion and exclusion criteria can be found in the manuscript Methods section (lines 97-106).

2. Based on the inclusion and exclusion criteria, we excluded 93 videos. A total of 107 videos were included in further assessments. We mentioned this in the Results section (lines 169-171).

3. We have included comprehensive information about 107 included videos in the S1 Table, and we have also supplemented the exclusion reasons for 93 excluded videos in the S2 Table.

References

1.Maganur PC, Hakami Z, Raghunath RG, et al. Reliability of Educational Content Videos in YouTubeTM about Stainless Steel Crowns. Children (Basel). 2022; 9(4): 571.

2.Nason K, Donnelly A, Duncan HF. YouTube as a patient-information source for root canal treatment. Int Endod J. 2016; 49(12): 1194-1200.

3.Sezici YL, Gediz M, Dindaroğlu F. Is YouTube an adequate patient resource about orthodontic retention? A cross-sectional analysis of content and quality. Am J Orthod Dentofacial Orthop. 2022; 161(1): e72-e79. 

4.Topsakal KG, Duran GS, Görgülü S, et al. Is YouTubeTM an adequate source of oral hygiene education for orthodontic patients? Int J Dent Hyg. 2022; 20(3): 504-511.

(4) Concern of the reviewer: How you accessed the video quality? is there any gold standard?

Our response: We are sorry that we did not explain clearly how we accessed the video quality in our previous response letter. As there is still no gold standard for evaluating video quality, therefore, our study referred to several recent studies on information quality analysis of YouTube videos on oral related topics using the global quality score (GQS) [1,2] and usefulness score [3] to evaluate video quality. While video reliability was assessed using DISCERN [4] (A full explanation of the methods used to access video quality analysis can be found in the manuscript Methods section (lines 121-148)). While these are mature standards, we look forward to a gold standard to be developed for evaluating video quality in the future that will help improve video quality by monitoring health information on YouTube and other social networks.

References

1.Sezici YL, Gediz M, Dindaroğlu F. Is YouTube an adequate patient resource about orthodontic retention? A cross-sectional analysis of content and quality. Am J Orthod Dentofacial Orthop. 2022; 161(1): e72-e79.

2.Ustdal G, Guney AU. YouTube as a source of information about orthodontic clear aligners. Angle Orthod. 2020; 90(3): 419-424. 

3.Çapan BŞ. YouTube as a source of information on space maintainers for parents and patients. PLoS One. 2021; 16(2): e0246431.

4.Charnock D, Shepperd S, Needham G, et al. DISCERN: an instrument for judging the quality of written consumer health information on treatment choices. J Epidemiol Community Health. 1999; 53(2): 105-111.

Reviewer 3:

(1)introduction line 29- its not explanatory, patients frequently present to the dental clinic with bleeding gums while brushing their teeth or noticing blood in their mouth in the morning after waking up.

Our response: We would like to express our gratitude to the reviewer for pointing out this error, as it will enhance the precision of the sentence. We have revised it (introduction, lines 29-31) , and have appropriately cited the corresponding literature in the reference list (References 1).

(2) line 33- classification is 2017 not 2018.

Our response: We thank the reviewer for pointing out this error and we have revised the year for the new classification of periodontal and peri-implant diseases and conditions (introduction, lines 35), which was proposed in 2017 as said by the reviewer [1].

References

1.Caton JG, Armitage G, Berglundh T, Chapple IL, Jepsen S, Kornman KS, et al. A new classification scheme for periodontal and peri‐implant diseases and conditions–Introduction and key changes from the 1999 classification. Journal of clinical periodontology. 2018; 45 Suppl 20: S1-S8.

(3) what were your keywords for search?

Our response: We are grateful to the reviewer for bringing this matter to our attention, referring to the information quality analysis study of YouTube videos on oral related topics [1-3], we were performed on YouTube using the keyword "bleeding gums" from Google Trends (details about the search strategies are included in the Materials and Methods section of our article (lines 73-84)).

References

1.Maganur PC, Hakami Z, Raghunath RG, Vundavalli S, Jeevanandan G, Almugla YM, et al. Reliability of Educational Content Videos in YouTubeTM about Stainless Steel Crowns. Children (Basel). 2022; 9(4): 571.

2.Sezici YL, Gediz M, Dindaroğlu F. Is YouTube an adequate patient resource about orthodontic retention? A cross-sectional analysis of content and quality. Am J Orthod Dentofacial Orthop. 2022; 161(1): e72-e79. 

3.Topsakal KG, Duran GS, Görgülü S, et al. Is YouTubeTM an adequate source of oral hygiene education for orthodontic patients? Int J Dent Hyg. 2022; 20(3): 504-511.

Thank you for your consideration. I look forward to hearing from you.

Best regards,

Prof. Minkui Lin

(Fujian Key Laboratory of Oral Diseases&Fujian Provincial Engineering Research Center of Oral Biomaterial&Stomatological Key Lab of Fujian College and University, School and Hospital of Stomatology, Fujian Medical University)

Email address: linmk105@ sina.com

---

## [Decision Letter · Decision Letter 2]

15 Jan 2024

PONE-D-23-16369R2YouTube as an information source for bleeding gums: A quantitative and qualitative analysisPLOS ONE

Dear Dr. Lin,

Thank you for submitting your manuscript to PLOS ONE. After careful consideration, we feel that it has merit but does not fully meet PLOS ONE’s publication criteria as it currently stands. Therefore, we invite you to submit a revised version of the manuscript that addresses the points raised during the review process.

Thank you for addressing the reviewers' comments. However, the manuscript needs further improvements. Kindly address all the comments made by the reviewers.==============================

We look forward to receiving your revised manuscript.

Kind regards,

Tanay Chaubal

Academic Editor

PLOS ONE

Reviewers' comments:

Reviewer's Responses to Questions

**Comments to the Author**

1. If the authors have adequately addressed your comments raised in a previous round of review and you feel that this manuscript is now acceptable for publication, you may indicate that here to bypass the “Comments to the Author” section, enter your conflict of interest statement in the “Confidential to Editor” section, and submit your "Accept" recommendation.

Reviewer #1: All comments have been addressed

Reviewer #3: All comments have been addressed

2. Is the manuscript technically sound, and do the data support the conclusions?

Reviewer #1: Yes

Reviewer #3: Yes

3. Has the statistical analysis been performed appropriately and rigorously? 

Reviewer #1: Yes

Reviewer #3: Yes

4. Have the authors made all data underlying the findings in their manuscript fully available?

Reviewer #1: Yes

Reviewer #3: Yes

5. Is the manuscript presented in an intelligible fashion and written in standard English?

Reviewer #1: Yes

Reviewer #3: Yes

6. Review Comments to the Author

Reviewer #1: Dear Author,

Rebuttal against the queries are not addressed satisfactorily. The manuscript is not acceptable in current form

Reviewer #3: All the queries and references are well explained and addressed to in proper systematic manner. all required questions have been answered and that all responses meet formatting specifications.

7. PLOS authors have the option to publish the peer review history of their article (what does this mean?). If published, this will include your full peer review and any attached files.

Reviewer #1: **Yes: **Akhilanand Chaurasia

Reviewer #3: No

---

## [Author Response · Author response to Decision Letter 2]

22 Jan 2024

Dear Editor:

Thank you for giving us the opportunity to submit a revised version of “YouTube as a Source of Information on Bleeding Gums: A Quantitative and Qualitative Analysis” (PLOS ONE manuscript number: PONE-D-23-16369) for potential publication in PLOS ONE. We appreciate your and the reviewers’ time and effort in providing feedback on our article. Your views, as well as the reviewers’, are valuable and have been instrumental in improving our research papers. We have carefully reviewed all of the suggestions and have made our best efforts to incorporate them into the text in order to meet the acceptance criteria. The following is our peer-to-peer response to these comments:

Reviewer Comments and Responses:

1.If the authors have adequately addressed your comments raised in a previous round of review and you feel that this manuscript is now acceptable for publication, you may indicate that here to bypass the “Comments to the Author” section, enter your conflict of interest statement in the “Confidential to Editor” section, and submit your "Accept" recommendation.

Reviewer #1: All comments have been addressed

Reviewer #3: All comments have been addressed

Our response: We express our gratitude to the reviewers for their endorsement.

2.Is the manuscript technically sound, and do the data support the conclusions? 

Reviewer #1: Yes

Reviewer #3: Yes

Our response: None.

3.Has the statistical analysis been performed appropriately and rigorously?

Reviewer #1: Yes

Reviewer #3: Yes

Our response: None.

4.Have the authors made all data underlying the findings in their manuscript fully available?

Reviewer #1: Yes

Reviewer #3: Yes

Our response: None.

5.Is the manuscript presented in an intelligible fashion and written in standard English?

Reviewer #1: Yes

Reviewer #3: Yes

Our response: None.

6.Review Comments to the Author

Reviewer #1: Dear Author,

Rebuttal against the queries are not addressed satisfactorily. The manuscript is not acceptable in current form.

We appreciate Dr. Akhilanand Chaurasia’s acknowledgement of our efforts to improve the manuscript based on the comments in the first round of review. However, there are still several areas that require further clarification. We have tried our best to review the manuscript and made sure that the updated responses below fully meet your requirements.

(1) What is novelty of your research ? 

Our response: According to our knowledge, no recent study has evaluated content related to bleeding gums on YouTube. However, study on YouTube video relevant to bleeding gums is required, as bleeding gums present a prevalent dental issue in individuals' everyday lives [1-3]. Bleeding gums are mainly triggered by periodontal disease and occasionally by peri-implant diseases, direct trauma, viruses, fungal or bacterial infections, medications, pregnancy, dermatoses, and systemic disorders [4-12], patients' quality of life suffers as a result [13-15].

The utilization of the YouTube platform for the purpose of seeking health-related information has become prevalent among individuals [16], and it is important to note that YouTube is open access, which means that user-generated content is not peer reviewed; while it offers effective health education resources, there may also be a great deal of misinformation and inaccuracy [17]. Incorrect and inadequate content in YouTube videos about bleeding gums can significantly influence patients' attitudes and medical decisions. Consequently, the objective of this study is to perform both quantitative and qualitative analyses of YouTube videos pertaining to bleeding gums.

In our article, we have directly stated the novelty of our research in the abstract (lines 11-13), introduction (lines 61-64), and discussion (lines 248-249). In addition, we added relevant references to the discussion, highlighting studies with both intra-oral and extra-oral causes, such as bad breath [18], with similar results [19], reinforcing our conclusions. At the same time, in order to better evaluate YouTube videos, we also refer to the literature of Chaurasia et al. [20], and add the possible role of Deep Learning model related content in YouTube video evaluation. Although it has not been realized in our study, Deep Learning model provides an important direction for future application.

References

1.Lawal FB, Dosumu EB. Self-reported and clinically evident gingival bleeding and impact on oral health-related quality of life in young adolescents: a comparative study. Malawi Med J. 2021; 33(2): 121-126.

2.Molina GF, Faulks D, Mazzola I, et al. Three-year survival of ART high-viscosity glass-ionomer and resin composite restorations in people with disability. Clin Oral Investig. 2018; 22(1): 461-467.

3.Chen H, Zhang R, Cheng R, et al. Gingival bleeding and calculus among 12-year-old Chinese adolescents: a multilevel analysis. BMC Oral Health. 2020; 20(1): 147.

4.Newbrun E. Indices to measure gingival bleeding. Journal of periodontology. 1996; 67(6): 555-561.

5.Darby I. Drugs and gingival bleeding. Australian Prescriber. 2006; 29: 154-155.

6.Bui FQ, Almeida-da-Silva CLC, Huynh B, Trinh A, Liu J, Woodward J, et al. Association between periodontal pathogens and systemic disease. Biomedical journal. 2019; 42(1): 27-35.

7.Rokaya D, Srimaneepong V, Wisitrasameewon W, Humagain M, Thunyakitpisal P. Peri-implantitis Update: Risk Indicators, Diagnosis, and Treatment. European journal of dentistry. 2020; 14(4): 672-682.

8.Heboyan A, Syed AUY, Rokaya D, Cooper PR, Manrikyan M, Markaryan M. Cytomorphometric Analysis of Inflammation Dynamics in the Periodontium Following the Use of Fixed Dental Prostheses. Molecules. 2020; 25(20): 4650.

9.Elani HW, Starr JR, Da Silva JD, Gallucci GO. Trends in Dental Implant Use in the U.S., 1999-2016, and Projections to 2026. Journal of dental research. 2018; 97(13): 1424-1430.

10.Lee CT, Huang YW, Zhu L, Weltman R. Prevalences of peri-implantitis and peri-implant mucositis: systematic review and meta-analysis. Journal of dentistry. 2017; 62: 1-12.

11.Achoki T, Miller-Petrie MK, Glenn SD, Kalra N, Lesego A, Gathecha GK, et al. Health disparities across the counties of Kenya and implications for policy makers, 1990-2016: a systematic analysis for the Global Burden of Disease Study 2016. Lancet Glob Health. 2019; 7(1): e81-e95.

12.Pihlstrom BL, Michalowicz BS, Johnson NW. Periodontal diseases. Lancet. 2005;366(9499):1809-1820.

13.Santonocito S, Palazzo G, Indelicato F, Chaurasia A, Isola G. Effects induced by periodontal disease on overall quality of life and self-esteem. Mediterranean Journal of Clinical Psychology. 2022; 10(1).

14.Slots J. Periodontology: past, present, perspectives. Periodontol 2000. 2013;62(1):7–19.

15.Frencken JE, Sharma P, Stenhouse L, Green D, Laverty D, Dietrich T. Global epidemiology of dental caries and severe periodontitis—a comprehensive review. Journal Of Clinical Periodontology. 2017;44(Suppl 18):S94-s105.

16.Bezner SK, Hodgman EI, Diesen DL, Clayton JT, Minkes RK, Langer JC, et al. Pediatric surgery on YouTube™: is the truth out there? Journal of Pediatric Surgery. 2014; 49(4): 586-589.

17.Hassona Y, Taimeh D, Marahleh A, Scully C. YouTube as a source of information on mouth (oral) cancer. Oral Dis. 2016; 22(3): 202-208.

18.Chaurasia A, Katheriya G. Prevalence of halitosis and related factors in North Indian population- A hospital based cross-sectional study. Journal of Oral Medicine, Oral Surgery, Oral Pathology and Oral Radiology. 2018; 4(2):79-84.

19.Ramadhani A, Zettira Z, Rachmawati YL, Hariyani N, Maharani DA. Quality and Reliability of Halitosis Videos on YouTube as a Source of Information. Dentistry Journal (Basel). 2021; 9(10): 120.

20.Chaurasia A, Namachivayam A, Koca-Ünsal RB, Lee JH. Deep-learning performance in identifying and classifying dental implant systems from dental imaging: a systematic review and meta-analysis. Journal of Periodontal and Implant Science. 2023; 53(3):e12.

(2) Concern of the reviewer: the methodology seems vague particularly study design? Clarify it. 

Our response: We appreciate the opportunity to explain our research method (especially the study design). Our research design is a cross-sectional study that quantitatively and qualitatively analyzes videos related to bleeding gums on YouTube. Within the framework of this research design, referring to the previous study conducted by Antonio et al [1-3], we provide a detailed description of the specific techniques and procedures used to collect, analyze, and interpret data in the Materials and Methods section. Our research method is as follows (The detailed research methods are included in the Materials and Methods section of our manuscript):

1. Conducted a search on YouTube using the keyword "bleeding gums" from Google Trends, select the top 200 videos, and further analyze 107 of them based on the inclusion and exclusion criteria;

2.Descriptive statistics on 107 videos, including the time since upload, the video length, and the number of likes, views, comments, subscribers, and viewing rates, and categorize these videos according to their sources;

3.Use global quality score (GQS), usefulness score, and total score to evaluate video quality, and use DISCERN to evaluate video reliability;

4.IBM SPSS Statistics 27 software was used for statistical tests. The normality of the data was determined using the Shapiro‒Wilk test. Non-parametric tests were used for data that did not adhere to the normal distribution: the Kruskal-Wallis test (for more than two samples), the Mann-Whitney test (for two independent samples), and Spearman's correlation coefficient. Kappa coefficients of correlation were calculated to assess interrater reliability. The significance level was set at p <0.05. 

References

1.Romano A, Lauritano D, Fiori F, et al. Cross-sectional study on the quality of oral lichen planus videos on YouTube™. J Oral Pathol Med. 2021; 50(2): 220-228.

2.Sezici YL, Gediz M, Dindaroğlu F. Is YouTube an adequate patient resource about orthodontic retention? A cross-sectional analysis of content and quality. Am J Orthod Dentofacial Orthop. 2022; 161(1): e72-e79. 

3.Uzel İ, Ghabchi B, Akalın A, et al. YouTube as an information source in paediatric dentistry education: Reliability and quality analysis. PLoS One. 2023; 18(3): e0283300.

(3) Concern of the reviewer: the inclusion and exclusion criteria also needs to be clarified. 

Our response: We are sorry that our previous response letter did not clearly explain the inclusion and exclusion criteria of our study, and the updated explanation is as follows:

1.Since the results may be heavily influenced by the inclusion and exclusion criteria, we referred to research on information quality analysis of YouTube videos on oral related topics [1-4] when formulating the inclusion and exclusion criteria, aiming to select research objects with common characteristics from complex videos and exclude the influence of some non-research factors. Ensuring the reasonableness and completeness of the inclusion and exclusion criteria will undoubtedly make the conclusions more solid and reliable. Detailed information on the inclusion and exclusion criteria can be found in the manuscript Methods section (lines 97-106).

2. Based on the inclusion and exclusion criteria, we excluded 93 videos. A total of 107 videos were included in further assessments. We mentioned this in the Results section (lines 169-171).

3. We have included comprehensive information about 107 included videos in the S1 Table, and we have also supplemented the exclusion reasons for 93 excluded videos in the S2 Table.

References

1.Maganur PC, Hakami Z, Raghunath RG, et al. Reliability of Educational Content Videos in YouTubeTM about Stainless Steel Crowns. Children (Basel). 2022; 9(4): 571.

2.Nason K, Donnelly A, Duncan HF. YouTube as a patient-information source for root canal treatment. Int Endod J. 2016; 49(12): 1194-1200.

3.Sezici YL, Gediz M, Dindaroğlu F. Is YouTube an adequate patient resource about orthodontic retention? A cross-sectional analysis of content and quality. Am J Orthod Dentofacial Orthop. 2022; 161(1): e72-e79. 

4.Topsakal KG, Duran GS, Görgülü S, et al. Is YouTubeTM an adequate source of oral hygiene education for orthodontic patients? Int J Dent Hyg. 2022; 20(3): 504-511.

(4) Concern of the reviewer: How you accessed the video quality? is there any gold standard?

Our response: We are sorry that we did not explain clearly how we accessed the video quality in our previous response letter. As there is still no gold standard for evaluating video quality, therefore, our study referred to several recent studies on information quality analysis of YouTube videos on oral related topics. Global quality score (GQS) [1,2] and usefulness score [3] were respectively used to evaluate the quality of YouTube video information to improve the accuracy of quality analysis results. The results showed that the quality scores of videos using GQS and usefulness scores were consistent, respectively (A full explanation of the methods used to access video quality analysis can be found in the manuscript Methods section (lines 121-148)). While these are mature standards, we look forward to a gold standard to be developed for evaluating video quality in the future that will help improve video quality by monitoring health information on YouTube and other social networks.

References

1.Sezici YL, Gediz M, Dindaroğlu F. Is YouTube an adequate patient resource about orthodontic retention? A cross-sectional analysis of content and quality. Am J Orthod Dentofacial Orthop. 2022; 161(1): e72-e79.

2.Ustdal G, Guney AU. YouTube as a source of information about orthodontic clear aligners. Angle Orthod. 2020; 90(3): 419-424. 

3.Çapan BŞ. YouTube as a source of information on space maintainers for parents and patients. PLoS One. 2021; 16(2): e0246431.

Reviewer #3:

All the queries and references are well explained and addressed to in proper systematic manner. all required questions have been answered and that all responses meet formatting specifications.

Our response: We thank Reviewer #3 for recognition of our revised manuscript. 

Thank you for your consideration. I look forward to hearing from you.

Best regards,

Prof. Minkui Lin

(Fujian Key Laboratory of Oral Diseases&Fujian Provincial Engineering Research Center of Oral Biomaterial&Stomatological Key Lab of Fujian College and University, School and Hospital of Stomatology, Fujian Medical University)

Email address: linmk105@ sina.com

---

## [Decision Letter · Decision Letter 3]

29 Jan 2024

YouTube as an information source for bleeding gums: A quantitative and qualitative analysis

PONE-D-23-16369R3

Dear Dr. Minkui Lin,

We’re pleased to inform you that your manuscript has been judged scientifically suitable for publication and will be formally accepted for publication once it meets all outstanding technical requirements.

Kind regards,

Tanay Chaubal

Academic Editor

PLOS ONE

Additional Editor Comments (optional):

Reviewers' comments:

Reviewer's Responses to Questions

**Comments to the Author**

1. If the authors have adequately addressed your comments raised in a previous round of review and you feel that this manuscript is now acceptable for publication, you may indicate that here to bypass the “Comments to the Author” section, enter your conflict of interest statement in the “Confidential to Editor” section, and submit your "Accept" recommendation.

Reviewer #1: (No Response)

2. Is the manuscript technically sound, and do the data support the conclusions?

Reviewer #1: (No Response)

3. Has the statistical analysis been performed appropriately and rigorously? 

Reviewer #1: (No Response)

4. Have the authors made all data underlying the findings in their manuscript fully available?

Reviewer #1: (No Response)

5. Is the manuscript presented in an intelligible fashion and written in standard English?

Reviewer #1: (No Response)

6. Review Comments to the Author

Reviewer #1: Dear Author,

Thank you for your efforts for elaborative explanation of queries raised. The rebuttal against the queries are satisfactory with sufficient length of explanation

7. PLOS authors have the option to publish the peer review history of their article (what does this mean?). If published, this will include your full peer review and any attached files.

Reviewer #1: No

---

## [Editor Report · Acceptance letter]

26 Feb 2024

PONE-D-23-16369R3 

PLOS ONE

Dear Dr. Lin, 

I'm pleased to inform you that your manuscript has been deemed suitable for publication in PLOS ONE. Congratulations! Your manuscript is now being handed over to our production team.

Kind regards, 

on behalf of

Dr. Tanay Chaubal 

Academic Editor

PLOS ONE